# Identification of circRNAs Associated with Adipogenesis Based on RNA-Seq Data in Pigs

**DOI:** 10.3390/genes13112062

**Published:** 2022-11-07

**Authors:** Qiaowei Li, Liyuan Wang, Kai Xing, Yalan Yang, Adeyinka Abiola Adetula, Yuwen Liu, Guoqiang Yi, Hongfu Zhang, Torres Sweeney, Zhonglin Tang

**Affiliations:** 1Kunpeng Institute of Modern Agriculture at Foshan, Foshan 528200, China; 2Shenzhen Branch, Guangdong Laboratory for Lingnan Modern Agriculture, Agricultural Genomics Institute at Shenzhen, Chinese Academy of Agricultural Sciences, Shenzhen 518124, China; 3School of Veterinary Medicine, University College Dublin, Belfield, D04 V1W8 Dublin, Ireland; 4Key Laboratory of Livestock and Poultry Multi-Omics of MARA, Agricultural Genomics Institute at Shenzhen, Chinese Academy of Agricultural Sciences, Shenzhen 518124, China; 5Innovation Group of Pig Genome Design and Breeding, Research Center for Animal Genome, Agricultural Genomics Institute at Shenzhen, Chinese Academy of Agricultural Sciences, Shenzhen 518124, China; 6Research Centre of Animal Nutritional Genomics, State Key Laboratory of Animal Nutrition, Agricultural Genomics Institute at Shenzhen, Chinese Academy of Agricultural Sciences, Shenzhen 518124, China; 7Animal Science and Technology College, Beijing University of Agriculture, Beijing 102206, China

**Keywords:** circRNAs, adipogenesis, network, RNA-seq, pigs

## Abstract

Adipocytes or fat cells play a vital role in the storage and release of energy in pigs, and many circular RNAs (circRNAs) have emerged as important regulators in various tissues and cell types in pigs. However, the spatio-temporal expression pattern of circRNAs between different adipose deposition breeds remains elusive. In this study, RNA sequencing (RNA-seq) produced transcriptome profiles of Western Landrace (lean-type) and Chinese Songliao black pigs (obese-type) with different thicknesses of subcutaneous fat tissues and were used to identify circRNAs involved in the regulation of adipogenesis. Gene expression analysis revealed 883 circRNAs, among which 26 and 11 circRNAs were differentially expressed between Landrace vs. Songliao pigs and high- vs. low-thickness groups, respectively. We also analyzed the interaction between circRNAs and microRNAs (miRNAs) and constructed their interaction network in adipogenesis; gene ontology classification and pathway analysis revealed two vital circRNAs, with the majority of their target genes enriched in biological functions such as fatty acids biosynthesis, fatty acid metabolism, and Wnt/TGF-β signaling pathways. These candidate circRNAs can be taken as potential targets for further experimental studies. Our results show that circRNAs are dynamically expressed and provide a valuable basis for understanding the molecular mechanism of circRNAs in pig adipose biology.

## 1. Introduction

Pigs are important farm animals that provide nutrition and energy for humans and are remarkable for biomedical research and applications. Pigs are highly similar to humans in terms of physiology, biochemistry, obesity, and diabetes, so they are one of the most suitable animal models for human biology and disease research [1]. Adipose tissue is a perplexing and significant metabolic and endocrine organ that develops mainly from precursor fat cells [2]. It not only plays an essential role in energy storage but also participates in nutrient homeostasis and regulates energy mobilization [3,4]. In pigs, the thickness of subcutaneous adipose is an important production trait closely related to meat mass and quality. The thickness of subcutaneous adipose has a positive correlation with intramuscular adipose in the longissimus dorsi and gluteus medius muscles [5]. The process of adipose deposition shows great heterogeneity among different pig breeds, which results from different domestication mechanisms [6]. Many previous studies have indicated that Chinese indigenous pig breeds have different genetic backgrounds from Western commercial breeds, especially in adipose deposition traits [7,8]. It is well known that the Landrace pig is representative of the fast-growing lean type [9]. Conversely, Songliao black pig is a fatty Chinese breed. Recently, studies integrating different breeds and different phenotypes in each breed became an efficient way to investigate underlying mechanisms for complex traits [10]. Therefore, it is necessary to understand the mechanism of adipose deposition in different breeds for the improvement in production efficiency and meat quality. In the past few years, researchers have made progress in adipose deposition by analyzing mRNA and miRNA for backfat thickness in pigs [11,12,13]. miR-224 was reported to have an important role in regulating fatty acid metabolism [14]. miR-100-5p, miR-143-3p, miR-10B-5p, and miR-24-3p were reported to be involved in fatty acid metabolism and fatty acid biosynthesis [15]. Several genes were discovered including *ADSSL1*, *AKAP5*, *BCAT1*, *HMOX1*, *PLIN2*, *ACP5*, *BCL2A1*, and *CCR1*, which were related to adipose metabolic pathways in large white pigs [12]. Through co-expression network analysis of weighted genes, *RAD9A*, *IGF2R*, *SCAP*, *TCAP*, *SMYD1*, *PFKM*, *DGAT1*, *GPS2*, *IGF1*, *MAPK8*, *FABP*, *FABP5*, *LEPR*, *UCP3*, *APOF*, and *FASN* were associated with fat deposition in Landrace and Songliao black pigs [16]. Furthermore, Liu et al. reported that adipose deposition and metabolism were associated with adipocytokines, calcium signaling, MAPK, FOXO, and PI3k/Akt signal pathways in lean-type Duroc and obese-type Luchuan pigs [17]. More detailed and integrated analysis is necessary to reveal the molecular mechanism of adipogenesis in pigs.

Circular RNAs (circRNAs) are a class of significant single-stranded RNA molecules with a covalently closed loop structure formed by back-splicing, which were first uncovered in plant viroids [18] and thought to be the result of splicing errors [19]. At present, emerging evidence indicates that circRNAs are broadly expressed in eukaryotes [20,21,22,23] and show cell-type-, tissue- and developmental-stage-specific expression patterns [24,25]. Hanan et al. found that throughout the porcine brain development, circRNAs were evolutionarily conserved, highly expressed and dynamically modulated [26]. In our previous study, our findings demonstrated that circFgfr2 could regulate myogenesis and muscle regeneration via a feedback loop [27]. A total of 147 tissue-specific, 149 tissue-enriched and 371 tissue-enhanced circRNAs in the fat tissue of swine also were detected by our previous research [28]. Importantly, circRNAs have been shown to participate in the regulation of various biological processes including fat deposition and adipocyte differentiation through distinct mechanisms. For instance, under the condition of inhibition of circ-PLXNA1, the differentiation of duck adipocyte would be limited [29]. Two circRNAs (chr19:45387150|45389986bp and chr21:6969877|69753491bp), located in chromosomes 19 and 21, distributed 45387150-45389986 and 45387150-45389986 in the genome, were revealed to have a strong correlation with fat deposition associated genes in Chinese Buffalo (*Bubalus bubalis*) [30]. A recently published report discovered that the expression levels of circ*ARMH1* in chicken were significantly and negatively related to *PPAR* signaling pathway genes in abdominal adipocytes, such as peroxisome proliferator-activated receptor γ (*PPARγ*), fatty acid binding protein 4 (*FABP4*), perilipin 2 (*PLIN2*), and CD36 molecule (*CD36*), while circ*CLEC19A* was significantly and positively related to those of genes [31]. In addition, chicken circLCLAT1 was detected with highly conservative characteristics among humans, mice, and pigs, which indicates that circLCLAT1 potentially regulates animal fat deposition or lipid metabolism [31]. In bovine, circFUT10 promotes adipocyte proliferation and inhibits adipocyte differentiation via sponging let-7 [32]. Liu et al. discovered that circSAMD4A regulated preadipocyte differentiation by acting as a miR-138-5p sponge and then increased EZH2 expression in humans [33]. In recent years, researchers found that adipose circRNAs showed dynamic regulation in obesity and functional role in fat deposition [34].

Recent studies indicated that the expression pattern of circRNAs in pig adipose tissue was related to adipose metabolism [35]. In the circRNA study of subcutaneous adipose in Large White and Laiwu pigs, the findings indicated that circRNA_26852 might bind ssc-miR-874 to regulate the expression of target genes and regulate adipocyte differentiation and lipid metabolism [36]. In recent years, circRNA expression profiles were constructed in fat tissues or fat-related cells in humans [37], mice [38], and swine [39], which provide crucial information for subsequent studies on fat deposition. However, fat-related circRNAs from Landrace and Songliao black pigs are rarely studied. To further understand the role of circRNAs in porcine fat deposition, here, we carried out a systematic analysis of circRNAs in subcutaneous adipose from Landrace and Songliao black pigs with extreme backfat thickness using RNA-seq data. We constructed circRNA-miRNA-mRNA interaction networks and identified two differentially expressed circRNAs (DECs) (Sus_CCL3L1_0001 and sus-intergenic_001702) that are potentially associated with fat deposition. Our findings provide promising circRNAs for further functional studies in adipocyte differentiation and metabolism, which might be used as candidate biomarkers for pig breeding.

## 2. Materials and Methods

### 2.1. Data Sources

In this study, the pigs came from a part of our previous RNA sequencing (RNA-seq) data [16]. In brief, we selected 500 purebred Landrace and Songliao black sows from the same pig farm. They were all healthy and raised in the same environment. We measured the backfat thickness of each living pig and grouped high backfat thickness and low backfat thickness populations from each breed. The backfat thickness in high-fat groups was 2–3 times higher than that in low-fat groups. In Landrace pigs, the average backfat thickness was 5.76 ± 1.75 mm. The average backfat thicknesses of the high and low groups were 8.7 mm and 3.80 mm, respectively. For Songliao black pigs, the average backfat thickness was 16.2 mm, and the high backfat thickness is 23.3 mm while the low backfat thickness was 9.1 mm. Additionally, then, total RNA was extracted from fat tissues for transcriptome sequencing. Utilize Illumina Hiseq 2000 sequencing platform, 12 paired-end sequencing data were obtained from Songliao black (*n* = 6) and Landrace pigs (*n* = 6) subcutaneous fat tissue, which included 3 samples with high backfat thickness and 3 samples with low backfat thickness in these two breeds, respectively. Additionally, the raw fastq files from the RNA data were all deposited at NCBI under accession no. SRP035333 for Songliao black [40] and SRP04004 for Landrace pigs [41]. The raw data included high and low backfat thickness groups in both pig breeds.

### 2.2. Data Quality Control

FastQC software (http://www.bioinformatics.babraham.ac.uk/projects/fastqc/, accessed on 4 October 2018) in Trim Galore was used for quality control and calculating the proportion of Q20 and Q30. Next, the FASTX-Toolkit program (http://hannonlab.cshl.edu/fastx_toolkit/, accessed on 2 February 2010) was adopted to trim the adaptor sequences with 20 as the quality threshold in each sample data point and discard low-quality reads. High-quality (Q > 20) clean reads were retained and used for further analysis. The software BBmap (http://sourceforge.net/projects/bbmap/, accessed on 13 June 2020) was utilized to align the reads and then filter an assembly by contig coverage. These processes guaranteed that the sequencing quality was high enough and accurate to perform further analysis.

### 2.3. Identification and Differential Expression Analysis of Sus scrofa circRNAs

All reads were mapped onto the genome of Sus scrofa.11.1 from Ensembl 95 by the program BWA-MEM [42]. Before aligning the clean data to Sus scrofa.11.1, we constructed an index for it and used the program CIRI2 to identify unique circRNAs, which were supported by less than two junction reads [43,44]. With FcircSEC (v1.0.0) R package, full length circRNA sequence were extracted and classified [45]. EdgeR-package in R was used to analyze the circRNAs expression manner [46]. We analyzed the expression differences of circRNAs between Landrace and Songliao black breeds in high and low-backfat groups, respectively. At the same time, the expression differences in circRNAs between high and low backfat groups within the same breeds also were analyzed. The circRNAs with fold-change >2 or <0.5 and false discovery rate (FDR) < 0.05 were defined as significant DE circRNAs using a Student’s *t*-test.

### 2.4. Target Prediction, KEGG Pathway Analysis and Network Construction

Two most highly expressed circRNAs, sus_CCL3L1_0001 and sus-intergenic_001702 were overlapped in Songliao black high backfat vs. Landrace high backfat, Songliao black low backfat vs. Songliao black high backfat (SH-VS-LH & SL-VS-SH) and Landrace low backfat vs. Landrace high backfat (LL-VS-LH), Landrace low backfat vs. Songliao black low backfat (LL-VS-SL) comparison. The two significantly differentially expressed and validated circRNAs were chosen to predict miRNA binding sites using both microtar [47], RNAhybrid program [48] and using the database of DIANA mir-Path bioinformatics resources [49]. Furthermore, the circRNAs and miRNAs were classified according to KEGG functional annotations to distinguish signal pathways actively regulated by miRNA in adipose tissue. Then, we constructed the circRNA-miRNA-pathway interaction networks with Cytoscape software (version 3.8.1) [50].

### 2.5. Validation of DE circRNAs

To validate the back-splice junctions of sus-intergenic_001702 and sus_CCL3L1_0001, head-to-tail splicing was assayed by quantitative polymerase chain reaction after reverse transcription (qRT-PCR), with divergent primers and Sanger sequencing (Table 1) [51]. The divergent primer could specifically amplify the back-splice junctions comparing to the linear sequence of the host gene. To confirm whether the back-splice sequences indeed reflect a distinct circular structure, DNase-treated total RNA was incubated for 15 min at 37 ℃ with or without Rnase R that degrades linear transcripts. RNA was subsequently purified by phenol-chloroform extraction.

### 2.6. Statistical Analysis

One-Way ANOVA in SPSS17.0 was used to analyze data here. LSD and Dunnett’s test were used to determine the significance among breed groups [52]. All statistical data are presented as mean ± SEM; a significance level of 0.05 was used. Fold-change >2 or <0.5 and false discovery rate (FDR) < 0.05 were set as thresholds for DE circRNAs screening. All experiments were carried out at least three times until producible results.

## 3. Results

### 3.1. Identification of Ssc circRNAs

To comprehensively detect circRNAs related to fat deposition in pigs, we carried out circRNAs analysis in high backfat Landrace (LH), low backfat Landrace (LL), high backfat Songliao black (SH), and low backfat Songliao black (SL). On average 56,157,514 reads were obtained per sample in this study (Appendix A). Using CIRI2 software, we identified a total of 883 circRNAs (Appendix A) that were supported by at least two reads spanning the back-splice junction. These circRNAs were flanked by GT/AG splice sites with well-defined breakpoints, and the distance was not more than 100 kb (Figure 1A). We detected 84.25% (744 out of 883), 55.38% (489 out of 883) and 43.60% (385 out of 883) of known circRNAs that derived from pigcirNet database (http://lnc.rnanet.org/circ, accessed on 27 October 2022) [28], circAtlas database (v2.0) (http://circatlas.biols.ac.cn/, accessed on 27 October 2022) [53] and our previous study, respectively [54]. A total of 366 circRNAs were detected in all four datasets. Of the identified circRNAs, 15.01% (133 out of 833) were novel in this study, which expands the catalogue of pig circRNAs to a certain extent (Figure 1B, Appendix A).

### 3.2. Molecular Characteristics of Ssc circRNAs

In the present study, we found that circRNAs were widely distributed along all chromosomes; the histogram show the distribution of circRNAs in autosomal and circRNAs were mainly located on chromosome 1, 6, and 12 (Figure 2A). According to the genomic source, circRNAs were categorized into four subtypes: exonic circRNAs (ecircRNAs) were derived from exon sequence, circular intronic RNAs (ciRNAs) only include intron sequence, exon–intron circRNAs (EIciRNA) consisted of exon and intron sequences, and intergenic circRNAs [55,56,57]. In this study, 74.18% (655 out of 833), 15.18% (134/833), and 10.65% (94/833) of circRNAs were sourced from exonic, intergenic, and intronic regions, respectively (Figure 2B). Next, the length of circRNAs identified in this study was compared with our previous report [54] as well as the circAtlas database [53]. We unearthed that the circRNAs from this study and the circAtlas database have higher similarity, and their median lengths were 371 bp and 372 bp, respectively (Figure 2C). In addition, characteristics analysis demonstrated that the median of the exon number was 4 and most of the circRNAs were circularized by no more than three exons (Figure 2D). Notably, there was a positive correlation between the number of circRNAs and mRNAs from each chromosome (Pearson R = 0.71, *p* = 6.68 × 10^−4^ < 0.05, Figure 2E) indicating the biogenesis of circRNAs relationship with mRNA. Most of the host genes (68.38%) only generated one circRNA, and about 6% of genes produced more than five circRNAs (Figure 2F).

### 3.3. Differentially Expressed (DE) circRNAs between Landrace and Songliao Black Pigs

To understand the regulatory roles of circRNAs in fat deposition between different breeds, we compared the expression level of circRNAs from Landrace and Songliao black pigs (Appendix A). For the group with high backfat, we identified 11 circRNAs differentially expressed between Songliao black and Landrace pigs. Among them, six and five circRNAs were up-regulated and down-regulated in Songliao black pigs, respectively (Figure 3A). In the group with low backfat, seven up-regulated and eight down-regulated circRNAs in Songliao black pigs were revealed (Figure 3B).

### 3.4. Differentially Expressed circRNAs between High and Low Backfat Group within Breeds

Next, we analyzed the DECs between groups with low and high backfat in Landrace and Songliao black pigs, respectively. The results evidenced that the DECs in the low backfat groups significantly differed from those in the high backfat groups. There were three up-regulated and four down-regulated circRNAs when circRNAs from the LL group were compared with circRNAs from the LH group (Figure 3C). We discovered three down-regulated and one up-regulated circRNAs between the SL group and the SH group (Figure 3D). These DECs mainly arose from host genes *CHD7, RNF169,* and *PHIP*. 

### 3.5. Validation of DECs

For two significant DECs, we performed reverse transcription (RT–PCR) with convergent and divergent primers. Convergent primers could amplify the desired band from circRNAs and linear mRNAs in both cDNA and gDNA, while only cDNA could amplify the desired band using divergent primers (Figure 4A). To further confirm whether the band indeed reflects the circular structure or was simply due to linear concatemers derived from genomic tandem duplications and exon reshuffling, RNase R digesting was performed before RT–PCR as the resistance of circRNAs to RNase R digestion. After RNase R digestion, only divergent primers could amplify the bands as we expected, indicating a closed-loop structure of circRNAs, while their corresponding linear cognates were susceptible to exonucleolytic cleavage (Figure 4B). Overall, sus_CCL3L1_0001 and sus-intergenic_001702 detected in our research validated the reliability of our analysis.

### 3.6. Functional Analysis of Sus_Intergenic_001702 and Sus_CCL3L1_0001

In order to detect the functions of circRNAs in fat deposition, we focused on fat deposition in the high- and low-fat content of Songliao and Landrace groups, and the expression levels of the four groups are displayed with the heatmap (Figure 5A). Nine DE circRNAs overlapped by more than two comparisons. Among them, the two most highly expressed circRNAs, sus_CCL3L1_0001 and sus-intergenic_001702 overlapped in SH-VS-LH, SL-VS-SH, LL-VS-LH and LL-VS-SL groups.

Previous studies revealed that circRNAs regulate mRNAs expression as miRNA sponges [58]. Here, we mainly focused on the two DE circRNAs (sus-intergenic_001702 and sus_CCL3L1_0001), which were differentially expressed between Landrace and Songliao black pigs as well as between high and low backfat groups. Firstly, we predicted miRNA binding sites of the circRNAs with computational methods and constructed the circRNA-miRNA-mRNA networks (Figure 5B). Subsequently, we inferred the potential functions of the two DE circRNAs based on the mRNAs with binding sites for these miRNAs. We found that these two DE circRNAs bind to let-7 and miR-27b, which are potentially involved in the biological process for fatty acid biosynthesis, fatty acid metabolism, biosynthesis of unsaturated fatty acids, and biosynthesis of unsaturated fatty acids [59,60] (Appendix A).

## 4. Discussion

Songliao black and Landrace pigs are suitable models for studying fat deposition, and they show significant differences in many ways, such as production and meat quality traits, particularly in fatness. The Songliao black pigs exhibit advantages in fast growth rate and high backfat thickness, and they have good reproductive characteristics. The main features of Landrace pigs are a large number of litters and high lean meat rate. The backfat thickness between Songliao black and Landrace pigs has a significant difference, so studying Songliao black and Landrace pigs in terms of adipose deposition has great value. Recently, with the development of circRNAs analysis programs, a large number of circRNAs were unfolded in humans, animals and plants [61,62]. The discovery of these widely and highly expressed circRNAs has increased the potential impact of circRNAs on biological function. For example, Ji et al. detected 71,112, 77,812, and 56,769 full-length circRNAs from 17 human, 13 macaque, and 14 mouse tissue samples, respectively [63]. It is worth noting that a total of 139,643 human and 214,747 mouse circRNAs were identified from 171 full-length single-cell RNA-seq datasets [64]. Meanwhile, 6519 circRNAs full-length circRNAs were studied in rice (*Oryza sativa*) and they were conserved in another 46 plant species [65]. In recent years, scientists have made some discoveries and performed some analyses of circRNAs related to the deposition, proliferation and differentiation of fat. It was reported that circFLT1 sponges miR-93 to regulate the proliferation and differentiation of adipocytes in bovine [66]. Novel circPPARA was discovered to promote intramuscular fat deposition in pigs [39]. However, at present, the expression patterns of circRNAs associated with adipose deposition from Songliao black and Landrace pigs have not been comprehensively profiled.

In the present study, we identified 883 circRNAs, consisting of 750 previously reported and 133 novel circRNAs from Songliao black and Landrace pigs. Our present research extended the porcine circRNAs resources about fat deposition, and many of the novel candidates had not been identified in publicly available datasets previously. We detected 28 unique DE circRNAs from SH-VS-LH, SL-VS-SH, LL-VS-LH, and LL-VS-SL comparison groups, and 32% (9/28) were overlapped by at least two groups, which supported the reasonability of our design. sus_CCL3L1_0001 and sus-intergenic_001702 were both significantly differentially expressed in the SL-VS-SH group and the LL-VS-LH group, respectively, suggesting their potential roles in fat deposition. sus-intergenic_001702 and sus_CCL3L1_0001 are engaged in lipid metabolism-related pathways, which would help explain the underlying functions of circRNAs in porcine fat deposition.

The functions of circRNAs acting as miRNA sponges [58] drove us to construct a circRNA-miRNA-mRNA network. In the above analysis, sus_CCL3L1_0001 and sus_intergenic_001702 were found to be closely related to fat deposition. Therefore, we focused on these two circRNAs in this study. These two validated DECs were predicted to bind to numbers of miRNAs, including let-7, miR-27b, miR-133, and miR-29, miR-218. Let-7 and the miR-27b family are engaged in fatty acid degradation signaling pathways, supporting the previous study that revealed that let-7 and miR-27b were involved in adipocyte differentiation [59,60]. A recent study demonstrated that miR-133 and miR-29 are related to obesity and diabetes [67,68] and regenerating muscle decreases, indicating that miR-133 might lead to the ectopic development of brown adipocytes and may be related to energy expenditure [69]. miR-218-5p affects subcutaneous fat production by targeting a new swine adipose deposition candidate, ACSL [70]. Our results showed that sus_CCL3L1_0001 and sus-intergenic_001702 may function in fat deposition via let7, miR-27b, miR-133, or miR-29. However, the two candidate DECs and their targets need further analysis. In the future, we will study the regulatory mechanism of sus_CCL3L1_0001 and sus-intergenic_001702 in adipose accumulation.

## 5. Conclusions

Taken together, RNA-Seq analysis provides an extensive catalog of circRNAs from Songliao black and Landrace pigs. Two DECs (sus-intergenic_001702 and sus_CCL3L1_0001) were validated as candidates for adipocyte differentiation. Prediction of miRNA targets for the two DECs and the construction of a circRNA–miRNA–mRNA network revealed that sus-intergenic_001702 and sus_CCL3L1_0001 were mainly engaged in fatty acids biosynthesis, fatty acid metabolism, and biosynthesis of unsaturated fatty acids pathways, suggesting their important roles in regulating adipocyte differentiation in pigs. This study provides the resources and foundation for the functional study of circRNAs in fat deposition.

## Figures and Tables

**Figure 1 genes-13-02062-f001:**
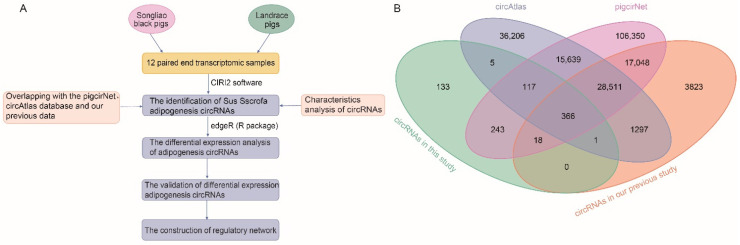
**Identification of *Ssc* circRNAs.** (**A**) The pipeline for circRNAs identification. (**B**) Venn diagram shows the intersection of circRNAs identified in this study (circRNAs in this study) with circRNAs from pigcircNet, circAtlas databases and circRNAs in our previous study.

**Figure 2 genes-13-02062-f002:**
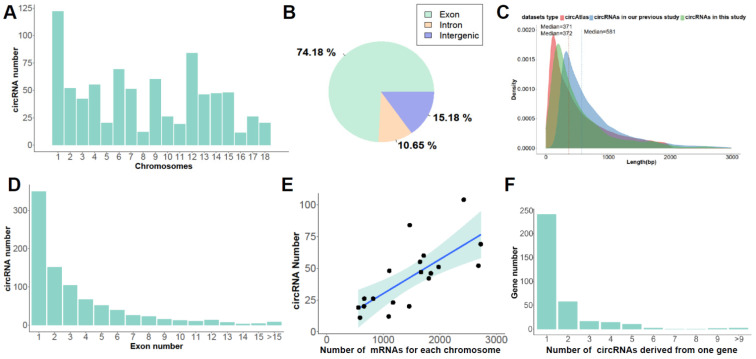
**Molecular characteristics of *Ssc* circRNAs in subcutaneous fat tissues.** (**A**) The distribution of circRNAs on the *Sus scrofa* chromosomes. (**B**) The distribution of circRNAs in genomic regions. (**C**) The comparison of circRNA length with other datasets. (**D**) Exon number of forming circRNAs. (**E**) The correlation between the number of circRNAs and the mRNAs for each chromosome. (**F**) The number of circRNAs generated from the host genes.

**Figure 3 genes-13-02062-f003:**
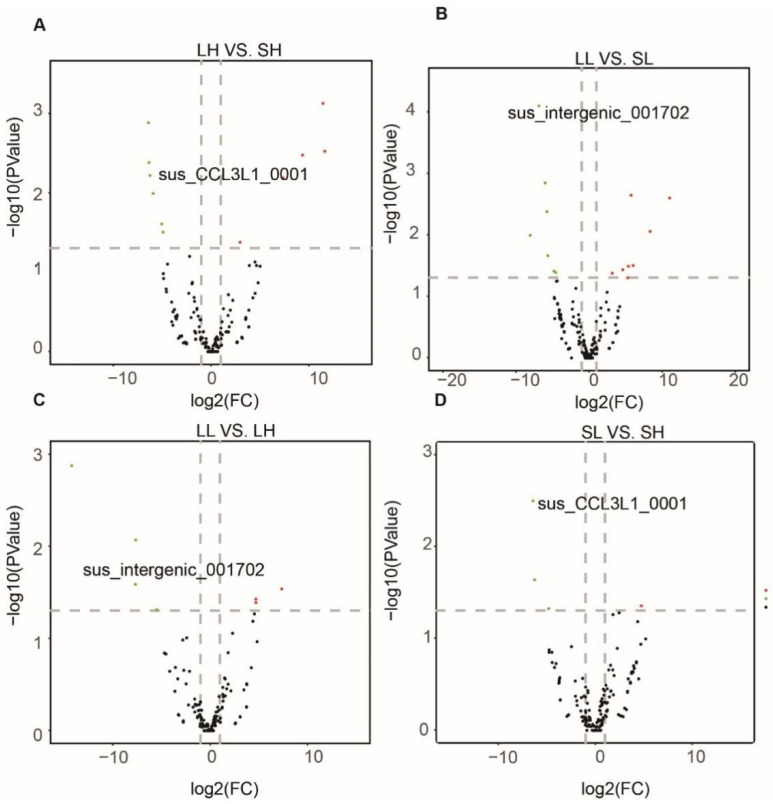
**The DECs between and within breeds.** (**A**) Significant DECs between LH and SH. (**B**) Significant DECs between LL and SL. (**C**) Significant DECs between low and high backfat of Landrace pigs. (**D**) Significantly DECs between low and high backfat of Songliao black pigs.

**Figure 4 genes-13-02062-f004:**
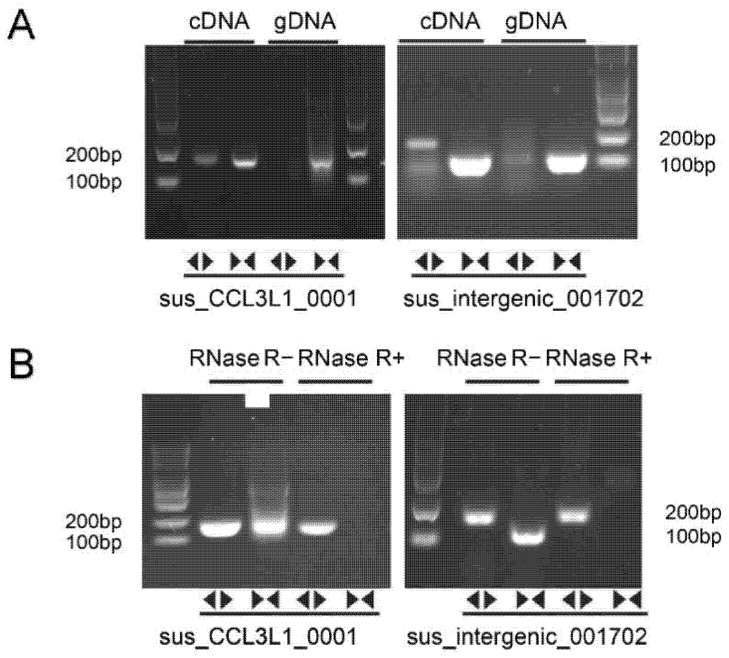
**PCR validation of DECs.** (**A**) PCR of circRNAs in cDNA and gDNA. (**B**) PCR of circRNAs after RNase R digestion. The first and third were convergent primers, while the second and fourth lane were divergent primers.

**Figure 5 genes-13-02062-f005:**
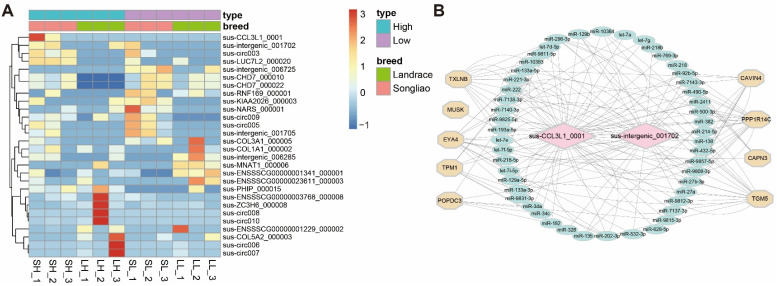
**The results of the functional analysis of the identified DECs.** (**A**) The heatmap of DE circRNAs in the four groups. (**B**) The circRNA–miRNA–mRNA interaction network.

**Table 1 genes-13-02062-t001:** Primers used for circRNA validation.

Primers	Sequence	Tm (°C)	Product Size (bp)
**sus_CCL3L1_0001 divergent primer**	TCAGCCTTACCTCTTCGC	58.2	185
	TGTGGGAGTGGACATCAGT		
**sus_CCL3L1_0001 convergent primer**	TTGGGCAAACACCTGAC	63.3	168
	CTGGCTCTGACCCTCTT		
**sus-intergenic_001702 divergent primer**	CACAAGGAGGAGGAAGCC	58.2	200
	AGGAGCCAGGTAAGAGCC		
**sus-intergenic_001702 convergent primer**	TGGTTGTCCGAGAAAGA	58.2	88
	GGTAGCTCGTCTGGTTT		

## Data Availability

Not applicable.

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
