# Peer review of "Identification of circRNAs Associated with Adipogenesis Based on RNA-Seq Data in Pigs"

_genes, 2022, doi:10.3390/genes13112062_

Round 1
Reviewer 1 Report (Previous Reviewer 4)
Dear Authors,
The performed analysis is interesting and provide valuable candidates for further studies regarding the mechanisms of adipogenesis. My main concern is the lack of the results of a part of the validation (Sanger and qPCR). Detailed suggestions are in the attachment.
Best wishes

Author Response
Dear reviewer1:
Thanks for your insightful comments and suggestions. According to your guidance, our revised manuscript has revised all of the wrong words, sentences, the content of the abscissa about Figure2 F, and incorrect places.
All the comments raised by you have been seriously considered. Please refer to the attachment for details.
Best wishes,
All authors

Reviewer 2 Report (Previous Reviewer 2)
Still problems with the english language to be solved. "Apposite models" in Discussion?
Author Response
Dear reviewer:
We agree with your suggestion and all the newly submitted manuscript have modified the terminology throughout the text as appropriate. And two native English authors (PhD. Adeyinka Abiola Adetula and Prof. Torres Sweeney) helped revised the language problems. And we revised the description as the “apposite models”.
Best wishes,
All authors
This manuscript is a resubmission of an earlier submission. The following is a list of the peer review reports and author responses from that submission.
Round 1
Reviewer 1 Report
see attached file

Reviewer 2 Report
Authors used RNA sequencing data from adipose tissue of two swine breeds for the identification of differentially expressed circular RNAs. According to the results they identified two differentially expressed circular RNAs whose expression is associated with fat deposition.
Compatibly with the absence of supplementary material, the strength is in the approach. Reference to individuals with strongly different phenotypes is certainly an advantage for identification of turning on/off of genes. The weakness is, probably, in the numbers. Reference to genetically related individuals characterised by different phenotypes would be certainly helpful. The construction of the interaction network is certainly complicated. I don’t know whether Authors could skip this point. However, if we consider costs and difficulties of interpretation of such an approach, I think that the effort can be appreciated.
In some cases, periods do not appear to be well constructed because of the absence of verbs. As far as I know, abbreviations can be used after that the form has been expressed in extenso. For the first time in my life, I had the possibility to read “Fascinatingly” in a scientific paper. I don’t even know whether this term exists in a correct english sense. I am sure that somebody able to correct minor errors in the writing form can be found, as well as of the fact that reviewers should not deal with the form of presentation of a manuscript. Editors should take care of this aspect
Please, use extended form the first time you use abbreviated items (miRNA).
Reviewer 3 Report
This manuscript is based on the differentel expression analysis of circular transcripts and considered 4 sets of 3 samples. This low number of samples to define a condition is clearly too low to propose a manuscript to Genes. Considering that 3 samples are enough to describe a condition, the authors consider that there is never an outlier sample.
before considering submitting this manuscript to another journal (with a lower IF), it should at least be demonstrated that 3 samples of each condition are close. Then it will also be necessary to propose a better quality analysis of circRNA. The flaws are numerous, the authors have only pushed buttons. Obviously they do not know the basic knowledge of these transcripts.
There is no originality in this manuscript. the scheme of this manuscript has already been proposed many times but in this manuscript the DE analysis is based on too few samples. Even for a journal with a lower impact factor, this manuscript cannot be considered for publication as proposed.
Reviewer 4 Report
Dear Authors,
The study is well designed and valuable. However, some inconsistencies have to be explained, and sentences in some parts reorganized to be more clear to a reader. Moreover, the discussion section, in my opinion, is too short and not comprehensive enough to highlight the value and significance of the obtained results. Mentioned associations of the identified circRNAs with adipose tissue processes are very superficial and general. I suggest looking deeper into the obtained miRNAs, and especially mRNAs (they are hardly discussed), to propose more strong connections of circRNAs and adipose tissue mechanisms to show their potential, and candidates and directions for further research. Detailed comments are in the pdf file.
Best wishes
